# Effects of Mindfulness-Based Interventions on Biomarkers and Low-Grade Inflammation in Patients with Psychiatric Disorders: A Meta-Analytic Review

**DOI:** 10.3390/ijms21072484

**Published:** 2020-04-03

**Authors:** Kenji Sanada, Jesus Montero-Marin, Alberto Barceló-Soler, Daisuke Ikuse, Marie Ota, Akihito Hirata, Akira Yoshizawa, Rieko Hatanaka, Montserrat Salas Valero, Marcelo Demarzo, Javier García Campayo, Akira Iwanami

**Affiliations:** 1Department of Psychiatry, Showa University School of Medicine, Tokyo 157-8577, Japan; ikuself3211731@yahoo.co.jp (D.I.); mariemarie1015@yahoo.co.jp (M.O.); msa473u@gmail.com (A.H.); masao-y@iris.ocn.ne.jp (A.Y.); rhatanaka0@gmail.com (R.H.); iwanami@med.showa-u.ac.jp (A.I.); 2Department of Psychiatry, University of Oxford, Warneford Hospital, Oxford OX3 7JX, UK; 3Primary Care Prevention and Health Promotion Research Network, RedIAPP, 28220 Madrid, Spain; abarcelosoler@hotmail.com (A.B.-S.);; 4Aragon Institute for Health Research, IIS Aragon, 50009 Zaragoza, Spain; 5Aragon Health Sciences Institute, IACS, 50009 Zaragoza, Spain; msalas.iacs@aragon.es; 6Mente Aberta-Brazilian Center for Mindfulness and Health Promotion, Department of Preventive Medicine, Universidade Federal de São Paulo, São Paulo 13565-905, Brazil; 7Instituto de Investigación Sanitaria Aragón, Hospital Universitario Miguel Servet, 50009 Zaragoza, Spain

**Keywords:** mindfulness-based interventions, biomarkers, low-grade inflammation, psychiatric disorders, meta-analysis

## Abstract

Mindfulness-Based Interventions (MBIs) present positive effects on mental health in diverse populations. However, the detailed associations between MBIs and biomarkers in patients with psychiatric disorders remain poorly understood. The aim of this study was to examine the effects of MBIs on biomarkers in psychiatric illness used to summarise the effects of low-grade inflammation. A systematic review of PubMed, EMBASE, PsycINFO, and the Cochrane Library was conducted. Effect sizes (ESs) were determined by Hedges’ g and the number needed to treat (NNT). Heterogeneity was evaluated. A total of 10 trials with 998 participants were included. MBIs showed significant improvements in the event-related potential amplitudes in attention-deficit hyperactivity disorder, the methylation of serotonin transporter genes in post-traumatic stress disorder, the salivary levels of interleukin 6 (IL-6) and tumour necrosis factor alpha (TNF-α) in depression, and the blood levels of adrenocorticotropic hormone (ACTH), IL-6, and TNF-α in generalised anxiety disorder. MBIs showed low but significant effects on health status related to biomarkers of low-grade inflammation (*g* = −0.21; 95% confidence interval (CI) –0.41 to −0.01; NNT = 8.47), with no heterogeneity (*I*^2^ = 0; 95% CI 0 to 79). More trials are needed to establish the impact of MBIs on biomarkers in psychiatric illness.

## 1. Introduction

Mindfulness is a theoretical construct that comes from Eastern spiritual traditions, especially Buddhism, and that has been recently adapted to Western culture and science. The word “mindfulness” is the English translation of *sati* (Pali) or *sm**ṛ**iti* (Sanskrit) which implies “bare attention” or “present-centred awareness” [1]. Kabat-Zinn introduced this therapeutic approach in the field of health in the 1970s [2], and defining mindfulness as the state of mind that arises from paying attention in a particular way, on purpose, in the present moment and without judgement [3]. He developed Mindfulness-Based Stress Reduction (MBSR) as a group treatment, reporting its effectiveness in patients with chronic pain [4]. Since then, new standardised intervention approaches based on MBSR have been developed, such as Mindfulness-Based Cognitive Therapy (MBCT), which aims to prevent relapses in depression [5], and Mindfulness-Based Relapse Prevention, which reduces the probability and severity of relapse in patients with substance use disorders [6,7]. These programmes are described as Mindfulness-Based Interventions (MBIs), with specific psychoeducational components adapted to the target population [1,8].

Although psychometric assessments of MBIs are generally limited, recent meta-analyses have shown the significant efficacy of MBIs for clinical symptoms of psychiatric disorders [9,10]; the authors reported that MBIs were equivalent to evidence-based treatments at post-treatment as well as at follow-up. Likewise, their positive effects on psychiatric symptoms have been found in patients with depression [11,12,13], anxiety [14,15,16], addictive disorders [17,18,19], insomnia [20,21], and somatoform disorder [22]. These previous studies indicate that mindfulness acts on different physiological markers related to anxiety by reducing cortisol, C-reactive protein (CRP), blood pressure, heart rate, triglycerides and tumour necrosis factor alpha (TNF-α). 

With regard to the scope of the study of mindfulness and its effects on neuroinflammation, the biomarkers that have been examined are the pro-inflammatory cytokines interleukin 6 (IL-6) [23,24,25,26], IL-8 [23,27], TNF-α [24,25,26] and interferon gamma (IFN-γ) [28,29,30]; the anti-inflammatory cytokines IL-4 [28] and IL-10 [24]; neuropeptides, such as neuropeptide Y [31]; and CRP [27,32]. In addition, although low-grade inflammation is generally determined by increased levels of high-sensitive CRP (hsCRP) [33,34], it is also reflected by increased concentrations of pro-inflammatory cytokines and acute phase proteins, such as CRP [35].

Despite the current evidence on mindfulness and biomarkers, the relationship between mindfulness and biomarkers in patients with psychiatric disorders remain poorly understood. Thus, the aim of this systematic review was to examine the effects of MBIs on biomarkers in patients with psychiatric disorders and to meta-analyse effects on low-grade inflammation from studies available to date.

## 2. Methods

This systematic review followed the Preferred Reporting Items for Systematic Reviews and Meta-Analyses (PRISMA) guidelines [36] and the recommendations of the Cochrane Collaboration [37]. The protocol was registered with the International Prospective Register of Systematic Reviews (PROSPERO) (registration number CRD42018084768).

### 2.1. Eligibility Criteria

The study’s eligibility criteria are shown in Table 1. We excluded studies for the following reasons: (1) inappropriate article type (i.e., study protocol) [38,39] and (2) inappropriate diagnostic procedure [40]. The articles that provided enough data to calculate effect size estimations of low-grade inflammatory biomarkers when comparing an MBI group with a control group were selected for the quantitative synthesis.

### 2.2. Search Strategy

A systematic computerised literature search of PubMed, EMBASE, PsycINFO and Cochrane Library was conducted by an expert in this field (M.S. Valero); in order to avoid language publication bias, there were no restrictions on language. As an example, the searching strategy for the PubMed database can be seen in Table 2. The reference lists of the identified original articles and reviews were also searched manually for additional studies that may have been missed. The last search was conducted on 26 November 2019.

### 2.3. Study Selection

Three authors (D. Ikuse, M. Ota and A. Yoshizawa) independently screened all titles and abstracts to identify possible articles for full text retrieval. The full texts of potentially eligible articles were assessed independently by the same three reviewers. Any discrepancies or divergences were resolved by discussion and consensus, and when in doubt, the final decision was made in consultation with a fourth author (K. Sanada).

### 2.4. Data Extraction

Using a predefined data extraction sheet, we extracted data for the following items: year of publication, characteristics of the groups and participants (i.e., number of participants, diagnoses of participants, age, gender), definition of psychiatric disorders, characteristics of MBIs and controls (type of intervention, length of intervention in total weeks), biomarkers (source, type) and study design.

### 2.5. Assessment of Study Quality

In the present systematic review, risk of bias was assessed with the Cochrane Collaboration tool [41] to assess possible sources of bias: sequence generation, allocation concealment, blinding of participants and personnel, blinding of outcome assessment, incomplete outcome data, selective outcome reporting and other sources of bias. The risk of bias tool is generally used for randomised controlled trials (RCTs), but it can also be applied to non-RCTs. Three reviewers (D. Ikuse, M. Ota and A. Yoshizawa) independently assessed these biases, and any disagreements between the authors were resolved through discussion or consultation with a fourth reviewer (K. Sanada). The quality of interventions was assessed by means of three criteria [42]: 1) use of a treatment manual, 2) provision of therapy by specifically trained therapists, and 3) verification of treatment integrity during the study.

### 2.6. Data Synthesis

Measurements were collected from the outcomes of such indices as adrenocorticotropic hormone (ACTH)*,* cortisol (area under the curve or AUC, awakening response or CAR, and diurnal slope), cytokines (IL-6, IL-8 and TNF-α), nuclear factor enhancer of the kappa light chains of activated B cells (NF-kB), high-sensitive CRP (hsCRP) and epidermal growth factor (EGF). ACTH is a melanocortin peptide produced in response to inflammatory mediators, with higher levels indicating a worse health status [43]. AUC is one of the methods for analysing the overall secretion of cortisol over a specific time-period (e.g., a whole day) in endocrinologic studies, with a higher AUC output considered to be an indicator of worse health status [8,44,45]. CAR is defined as the change in cortisol concentration that occurs during the first hour after waking from sleep [46] and low CAR values have been associated with low health states such as fatigue and burnout [8,47]. The diurnal slope is a method for analysing cortisol concentrations focused on the diurnal cycle, in which levels of cortisol are high in the morning and low at night, so that higher morning levels and diurnal slope values are considered to indicate a better health status [8,48,49]. The IL-6, IL-8 and TNF-α cytokines are of a pro-inflammatory type, and thus the higher the levels, the worse the state of health [50,51,52]. NF-kB is a pro-inflammatory transcription factor in which higher values correspond to worse health status [53]. High-sensitive CRP (hsCRP) is more sensitive than standard CRP, and there are some relationships between hsCRP and cardiovascular disease [54], with higher levels indicating a worse health status. EGF has been related to stress and mood disorders [55,56], with decreased levels being associated with improvements in symptoms. 

We examined the post-treatment measurements of low-grade inflammation that were collected immediately after the intervention. The effect size (ES), indicating the differences between the two groups (MBI vs control), and 95% confidence intervals (CIs) were calculated. Hedges’ g was chosen as the ES measure, since the present analysis included studies with a small sample size, and this measure adjusts accordingly [57]. If necessary, combined outcomes were estimated using a pooled mean ES. We also converted Hedges’ g into the number needed to treat (NNT), according to Furukawa’s formula [58]. NNT indicates the number of participants who need to be treated in order to generate one additional, clinically significant, favourable change [59]. We tested heterogeneity using the *I*^2^ statistic and 95% CI, assuming a value of around 25% to indicate low heterogeneity, 50% to indicate moderate and 75% to indicate high heterogeneity [60]. We also calculated the *Q* statistic and the associated *p*-value. Publication bias was assessed initially through the construction of a funnel plot analysis [61]. Egger’s test was used to contrast the null hypothesis with biased absences [62], and Duval and Tweedie’s trim and fill procedure [63] provided the number of studies that were probably absent. The Begg and Mazumdar rank correlation test was also applied to test whether the adjusted and observed ESs differed significantly from each other [64]. All of the tests were two-sided and were performed with a significance level of *p* < 0.05, except for the bias-related tests, which were one-sided.

## 3. Results

### 3.1. Search Results

We identified 213 titles and abstracts from PubMed (12), EMBASE (167), PsycINFO (9), and the Cochrane Library (25). We also identified seven possible additional records through other sources. There were 183 records after duplicate removal, 168 of which were excluded because they did not meet the inclusion criteria (Figure 1). A total of 15 articles were retrieved for eligibility and possible inclusion in the database. Of these, we finally selected 10 studies that examined the efficacy of MBIs on biomarkers in patients with psychiatric disorders. The PRISMA flow chart is shown in Figure 1. In the included studies, seven used RCTs [23,26,65,66,67,68,69], the remaining three being open-label studies [70,71,72]. The majority of studies did not include a follow-up assessment time point (70%). Six studies focused on depression and/or anxiety disorder, or depressive symptoms [23,26,65,68,69,71], one on alcohol dependence [70], one on attention-deficit hyperactivity disorder (ADHD) [66], one on sleep disturbance [67] and the other on post-traumatic stress disorder (PTSD) [72]. Six studies were conducted in the United States (60%) and the others in Europe. The characteristics of the 10 studies included for qualitative synthesis are listed in Table 3.

### 3.2. Participants

The total number of participants across the included studies was 998 (641 women, 64%). These comprised 19 patients with alcohol dependence [70], 56 remitted patients with recurrent depression [65], 44 patients with ADHD [66], 49 patients with moderate sleep disturbance [67], 11 patients with major depressive disorder (MDD) [71], 64 participants with depressive symptoms [68], 166 patients with mild to moderate depression or anxiety [23], 177 patients with depression, anxiety or stress and adjustment disorders (and 320 healthy controls) [69], 22 patients with PTSD [72] and 70 patients with generalised anxiety disorder (GAD) [26]. Of the data available (*n* = 898), the mean age of the participants was 42.6 (SD = 14.3) years. As stated in the inclusion criteria, all included studies in this review used a dichotomous measure for psychiatric disorders, namely, a diagnostic interview (Diagnostic and Statistical Manual of Mental Disorders-Fourth Edition [DSM-IV], Structured Clinical Interview for DSM-IV [SCID], International Statistical Classification of Diseases and Related Health Problems-10 [ICD-10]) and/or a cut-off for clinical symptoms on a rating scale (Montgomery-Åsberg Depression Rating Scale [MADRS], Pittsburgh Sleep Quality Index [PSQI], Hamilton Depression Rating Scale [HAMD], Center for Epidemiologic Studies-Depression Scale [CES-D], Patient Health Questionnaire-9 [PHQ-9], Hospital Anxiety and Depression Scale-Depression [HADS-D], Hospital Anxiety and Depression Scale-Anxiety [HADS-A]).

### 3.3. Quality of Studies and Interventions

The risk of bias in the included studies is shown in Figure 2 and Figure 3. Of the 10 studies included for qualitative synthesis, only two [67,69] showed a “low risk” of bias for at least five items. Out of seven items, only one item, “selective outcome reporting”, was considered to have a “low risk of bias”. With regard to the quality of the interventions, the use of a treatment manual was reported in five trials, therapist training in all trials and treatment integrity in four trials. The corresponding detailed assessment is shown in Table 3.

### 3.4. Biomarkers

Of the articles reviewed, the biomarkers mainly came from blood samples while four studies used salivary [65,68,70] and electroencephalogram (EEG) [66] samples. The biomarkers included blood IL-6 in patients with depression or anxiety [23], GAD [26], and alcohol dependence [70], salivary IL-6 in subjects with depressive symptoms [68], salivary cortisol in remitted patients with recurrent depression [65] and in patients with alcohol dependence [70], blood cortisol in patients with GAD [26], CRP in patients with MDD [71], hsCRP in patients with depression or anxiety [23], blood TNF-α in patients with GAD [26], salivary TNF-α in subjects with depressive symptoms [68], ACTH in patients with GAD [26], EGF in patients with depression or anxiety [23], IL-8 in patients with depression or anxiety [23], leukocyte telomere length in patients with depression or anxiety [69], NF-kB in patients with sleep disturbance [67] and event-related potential (ERP) in patients with ADHD [66]. Of the included studies, IL-6 and cortisol were the most frequently examined (three studies), followed by CRP and TNF-α (two studies); one study [23] did not ultimately analyse the levels of plasma IL-6 due to test sensitivity issues.

### 3.5. Mindfulness-Based Interventions (MBIs)

For the most part, MBIs comprised two types: MBSR [26,72] and MBCT [65,66,71]. Four studies were conducted with modified versions of MBSR, namely, mindfulness meditation relapse prevention [70], mindful awareness practices (MAPs) [67] and mindfulness-based group therapy [23,69]; another was performed with a brief MBI [68]. In six studies, the intervention lasted eight weeks [23,26,65,69,70,71]. The interventions in the other four studies lasted 12 weeks [66], nine weeks [72], six weeks [67] and four weeks [68].

### 3.6. Effectiveness of MBIs on Biomarkers

There was a high heterogeneity of biomarkers and participants, with studies showing a low quality in general. As observed in Table 3, MBIs showed some effects on biomarkers in six studies [26,66,68,70,71,72].

Statistically significant findings were found in four trials [26,66,68,72]. One trial [66] examined the effectiveness of a 12-week MBCT intervention on ERP in 50 patients with ADHD. Of the participants, 26 patients were randomised to an MBCT group, and 24 to a wait-list group. ERP, including error-positivity (Pe), conflict monitoring (NoGo-N2) and inhibitory control (NoGo-P3), was evaluated by recording EEG concomitant with a standard visual continuous performance task (CPT-X). The authors reported that MBCT enhanced Pe and NoGo-P3 ERP amplitudes (*p* = 0.02 and *p* = 0.02, respectively); increased Pe amplitudes were correlated with a reduction in hyperactivity/impulsivity symptoms, and enhanced NoGo-P3 amplitudes were associated with an improvement in inattention symptoms. Another trial [68] investigated the effects of a four-week MBI on salivary IL-6 and TNF-α in 64 participants with depressive symptoms; 31 participants were randomly allocated to the mindfulness group and 33 participants to the contact control group. MBI predicted significantly lower levels of IL-6 and TNF-α at post-treatment (*p* < 0.001 and *p* = 0.013, respectively). Another trial [72] reported DNA methylation related with treatment response to a nine-week MBSR intervention in 22 veterans with PTSD. The authors categorised the participants as responders (*n* = 11) and non-responders (*n* = 11) on the basis of a 10 point or more reduction on a PTSD Checklist (PCL) for symptom severity. There were significant methylation changes related to treatment response in *FKBP5* intron 7 bin 2 after a nine-week MBSR (*p* = 0.013). The other trial [26] investigated the effects of an eight-week MBSR intervention on stress response to the laboratory-based Trier Social Stress Test (TSST) in 70 patients with GAD. Of these subjects, 42 were randomly allocated to an MBSR group and 28 to the attention control group. The peripheral biomarkers, namely cortisol, ACTH, IL-6 and TNF-α, were measured at three TSST time periods: pre-stress, immediate post-stress and later post-stress. Statistically significant changes in the AUC concentrations for ACTH, TNF-α and IL-6 from baseline to post intervention were found in the MBSR group compared to the control group (*p* = 0.007, *p* = 0.033, *p* = 0.036, respectively).

Two open-label trials [70,71] reported benefits from MBIs in improving inflammatory biomarkers. Zgierska et al. [70] conducted an eight-week mindfulness meditation for 19 patients with alcohol dependence, with 16-week follow-up assessments of serum IL-6 and salivary cortisol. Although no significant changes in the levels of salivary cortisol from baseline to 16 weeks were found, serum IL-6 levels decreased, showing a trend at 16 weeks compared with baseline (*p* = 0.052), with a moderately high ES (Cohen’s *d* = 0.6). Eisendrath et al. [71] conducted an eight-week MBCT for 11 patients with MDD and assessed the levels of serum CRP at pre- and post-intervention. Serum CRP levels from pre- to post-intervention decreased, showing a trend (*p* = 0.052) with a moderate ES (Cohen’s *d* = 0.66).

By contrast, the other four RCTs [23,65,67,69] revealed no significant group differences in the respective effects of MBCT on salivary cortisol with one-year follow-up in remitted patients with recurrent depression compared to treatment as usual (TAU) [65]; of MAPs on plasma NF-kB in patients with sleep disturbance compared to an education intervention [67]; of mindfulness-based group therapy on plasma IL-8, EGF, and hsCRP in patients with depression or anxiety compared to cognitive behavioural therapy (CBT) [23]; or of mindfulness-based group therapy on leukocyte telomere length in patients with depression or anxiety compared to CBT [69].

With regard to the individual biomarkers, IL-6 and cortisol were the most frequently reported, each in three articles ([26,65,68,70], respectively). With regard to IL-6, two studies [26,70] measured serum or plasma IL-6 levels, and the other used salivary samples [68]. One study observed that the levels of serum IL-6 in patients with alcohol dependence did not significantly decrease although they showed a trend from baseline to 16-week follow-up (*p* = 0.052) [70]. The other two studies reported that the levels of salivary IL-6 in an MBI group with depressive symptoms significantly decreased within the group between baseline and post-treatment (*p* < 0.05) [68], and the levels of plasma IL-6 AUC concentration in an MBSR group with GAD significantly decreased compared to an education group during the TSST (*p* = 0.036) [26]. For cortisol, two studies used salivary samples [65,70] and the other used plasma samples [26]. There were no significant changes from baseline to the end of intervention or to follow-up in the levels of salivary cortisol in patients with alcohol dependence (*p* > 0.05) [70], or in remitted patients with recurrent depression [65]. In addition, no significant change in plasma cortisol AUC concentrations during the TSST was found between the MBSR group and the education group in patients with GAD (*p* = 0.38) [26].

With regard to CRP, the findings of two studies were consistent: there were no significant changes in the levels of CRP or hsCRP from baseline to post-intervention in patients with MDD or with depression or anxiety [23,71]. 

### 3.7. Low-Grade Inflammation Outcomes Synthesis

The data synthesis was composed of outcomes from five studies that included comparisons with control groups [23,26,65,67,68], having a total of 405 participants (206 in the MBI groups and 199 in the control groups) and using the following biomarkers: EGF, ACTH, cortisol, IL-6, IL-8, TNF-α, NF-kB and hsCRP. Figure 4 shows the Forest Plot for the overall ES found. As can be seen, MBIs showed low but significant effects in improving the state of health related to low-grade inflammation outcomes (*g* = −0.21; *p* = 0.043; NNT = 8.47), with no heterogeneity (*I*^2^ = 0; 95% CI = 0–79; *p* = 0.976). No indication of publication bias was found in the overall estimate (Begg tau = 0.40, *p* = 0.231; Egger intercept −0.22, *p* = 0.389). Duval and Tweedie’s trim and fill procedure did not propose the addition of new imputed studies. Because of the limited number of studies, a post-hoc power calculation was conducted to examine whether a sufficient number of studies and sample sizes had been taken into consideration in order to identify relevant effects. These calculations indicated that the inclusion of five studies, with a mean sample size of 41 participants per condition, a low degree of heterogeneity assumed and a significance level of α = 0.05, resulted in a statistical power of 0.47, in order to detect an ES of −0.21. 

## 4. Discussion 

In this first comprehensive meta-analytic review of the effects of MBIs on biomarkers in patients with psychiatric disorders, it was found that while several types of biomarkers and participants were included and the selected articles were of poor quality, significant findings were observed for MBIs in terms of improvements in the ERP amplitudes in patients with ADHD [66], in the methylation of serotonin transporter genes in patients with PTSD [72], in the salivary levels of IL-6 and TNF-α in participants with depressive symptoms [68] and in the blood levels of ACTH, IL-6, and TNF-α in patients with GAD [26]. 

With regard to ERP amplitudes, previous studies have generally reported a reduction of P3 amplitudes in adolescents with ADHD compared to control children [73,74,75]. A recent systematic review noted that MBIs could promote a positive relationship between emotional appraisal of errors and Pe in pre-adolescents [76]. Very few studies have been conducted on the effects of MBIs on ERP amplitudes in adult patients with ADHD. Thus, further studies are needed to examine the effects of MBIs on ERP and, specifically, on whether they have effects in adult patients as well as in adolescent or pre-adolescent patients with ADHD. 

With regard to DNA methylation biomarkers, FK506 binding protein 5 (*FKBP5*) regulates glucocorticoid receptor signalling by decreasing ligand binding and restricting its translocation to the nucleus [77,78]; the *FKBP5* gene could be associated with an increased number of lifetime depressive episodes [79]. In line with our results and those of Bishop et al. [72], one study demonstrated that the methylation of *FKBP5* promoter decreased in veterans with PTSD who responded to exposure psychotherapy [80]. However, there were no other studies that examined the effects of psychotherapy, including MBIs, on the methylation of the *FKBP5* gene in patients with PTSD. Thus, a larger number of studies are needed to clarify the association between *FKBP5* gene methylation and symptom severity in this population.

With regard to inflammatory cytokines, a recent meta-analysis found no significant effects of MBIs on the blood levels of IL-6 and TNF-α in participants with ulcerative colitis or obesity, or in dementia caregivers and healthy subjects [16]. Likewise, two previous systematic reviews reported that null findings of replicated effects of MBIs were observed for inflammatory cytokines in blood, including IL-6 and TNF-α, in participants with cancer, ulcerative colitis or rheumatoid arthritis, or in dementia caregivers and healthy controls [1,81]. For ACTH, one study performed by Kim et al. [82] found no significant effects from mindfulness-based stretching and deep breathing exercises based on MBSR on the levels of blood ACTH in nurses with subclinical features of PTSD. However, our findings revealed the positive effects of MBSR training on AUC concentrations for blood ACTH, IL-6 and TNF-α during the TSST in patients with GAD [26], and of a brief MBI on the levels of salivary IL-6 and TNF-α in participants with depressive symptomatology [68]. This discrepancy may in part be attributable to 1) the inadequate statistical power in terms of studies and sample sizes found, and 2) the various types of participants without psychiatric diseases. 

We also conducted a meta-analysis to examine the specific effects of MBIs on biomarkers related to low-grade inflammation. We observed low but significant effects in improving the state of health in patients with psychiatric disorders, including MDD, sleep disorder and anxiety disorders. Low-grade inflammation is associated with several major psychiatric disorders, such as MDD, anxiety disorders and schizophrenia. In a previous study [35], the prevalence rates for low-grade systematic inflammation, which was defined by serum CRP > 3mg/L, in unipolar mood disorder and in neurotic disorders, including anxiety disorders, were 16% and 22%, respectively. As mentioned earlier, in this review, we treated low-grade inflammation as the comprehensive involvement of the hypothalamic-pituitary-adrenal axis and pro-inflammatory cytokines (Figure 5) and included biomarkers related to low-grade inflammation, such as EGF, ACTH, cortisol, pro-inflammatory cytokines (IL-6, IL-8, TNF-α), NF-kB and hsCRP. In line with our results for CRP and hsCRP, a recent review from two RCTs concluded that MBIs showed no significant reductions in the inflammatory biomarker CRP in stressed community subjects [32] (this population did not meet our inclusion criteria). However, very few studies have explored the associations between MBIs and biomarkers of low-grade inflammation. Thus, further studies are needed to clarify the extent to which MBIs affect low-grade inflammation in patients with psychiatric disorders and in healthy subjects.

There are several limitations to the present meta-analytic systematic review. First, our quantitative synthesis has low statistical power due to the scarce number of studies and sample sizes. Therefore, more studies are clearly needed to obtain a powerful analysis. Second, the included studies had various types of biomarkers and patients with psychiatric disorders, and sub-group analyses could not be carried out due to the insufficient number of studies. Third, the included studies had a great diversity of control groups: waiting list, education, TAU and CBT. Fourth, most of the studies showed a high risk of bias; thus, studies of higher quality are needed. Fifth, the quality of interventions in most studies was rather low. Despite the limitations mentioned, this kind of work could help to overcome those shortcomings linked to the only use of self-report measures of trait mindfulness when using MBIs to treat psychiatric disorders, proposing possible biomarkers that could function as mechanisms of action in this specific type of interventions and populations.

In conclusion, our findings indicate that MBIs seems to be associated with significant improvements in the ERP amplitudes in patients with ADHD, in the methylation of serotonin transporter genes in patients with PTSD, in the salivary levels of IL-6 and TNF-α in participants with depressive symptoms and in the blood levels of ACTH, IL-6 and TNF-α in patients with GAD. In addition, MBIs seems to be related to a low but significant improvement in health status related to biomarkers of low-grade inflammation. In light of the small ES observed; however, further studies are required to improve the understanding of the effects of MBIs on biomarkers in psychiatric disorders.

## Figures and Tables

**Figure 1 ijms-21-02484-f001:**
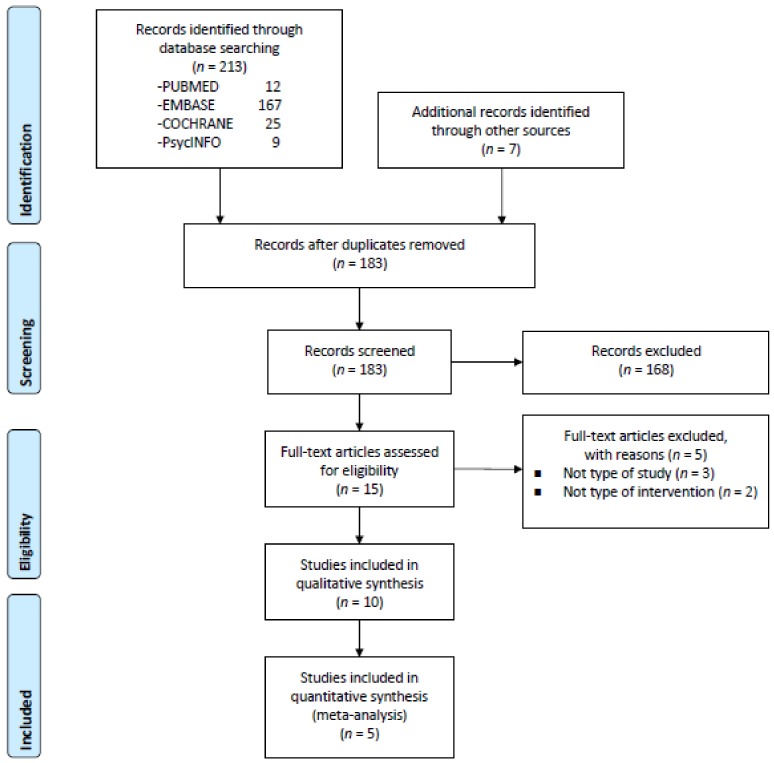
PRISMA flow chart of study selection.

**Figure 2 ijms-21-02484-f002:**
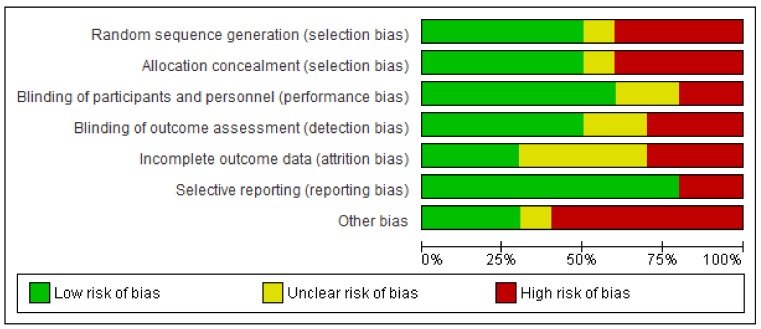
Risk-of-bias graph: reviews the authors’ judgments about each risk-of bias item presented as percentages across all of the included studies.

**Figure 3 ijms-21-02484-f003:**
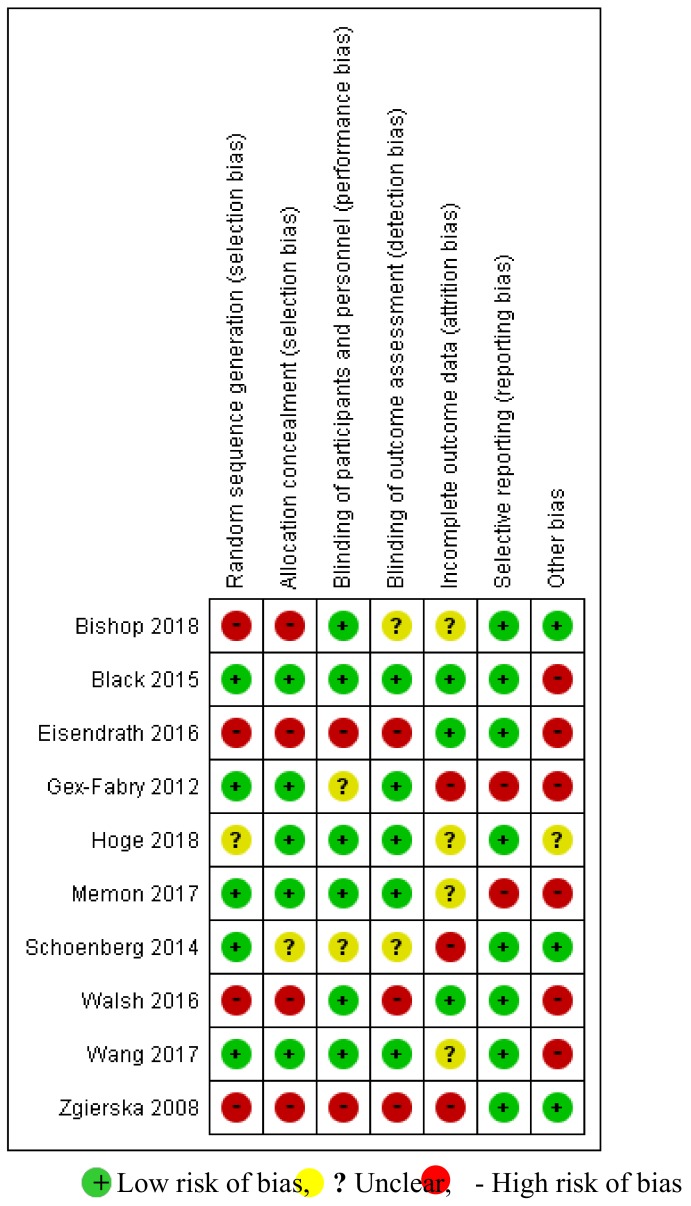
Risk-of-bias summary: review of the authors’ judgments about each risk-of-bias item for each included study.

**Figure 4 ijms-21-02484-f004:**
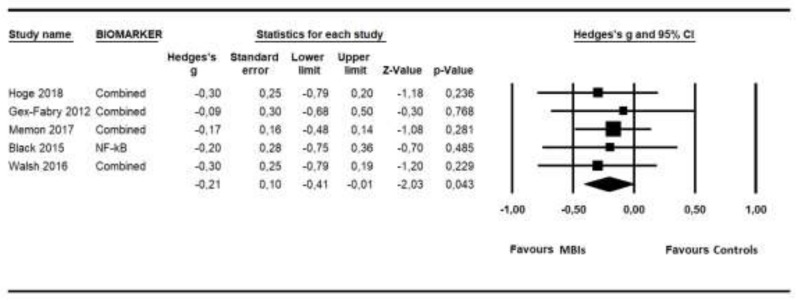
Forest Plot for the overall effect size of low-grade inflammatory biomarkers.

**Figure 5 ijms-21-02484-f005:**
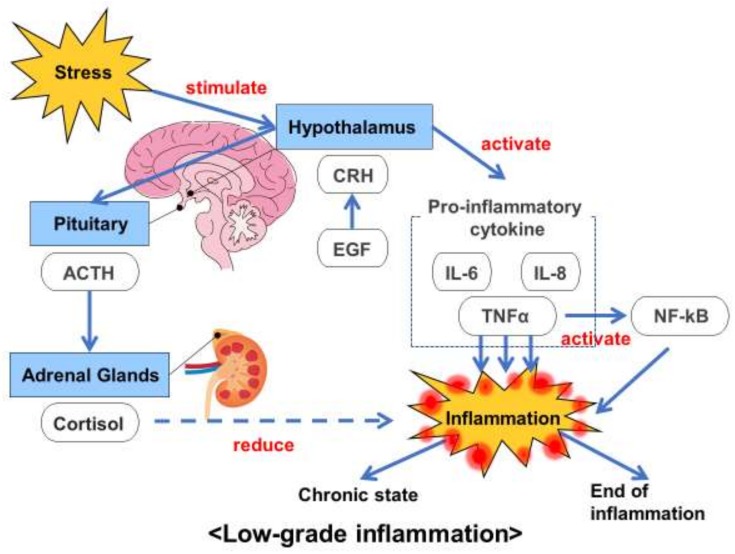
Association between biomarkers of low-grade inflammation and stress.

**Table 1 ijms-21-02484-t001:** Study eligibility criteria.

	Inclusion Criteria	Exclusion Criteria
**Participants**	Patients with psychiatric disorders according to either a formal diagnosis interview and/or a cut-off for clinical symptoms in a rating scale; No restrictions were placed on age.	Patients with other disorders, and only healthy subjects.
**Interventions**	Mindfulness-based interventions (MBIs)	Other non-pharmacological interventions.
**Outcome**	At least one biomarker.	No biomarkers.
**Study design**	RCTs, Non-RCTs, Open trials with a pre-post analysis.	Study protocols, cross-sectional studies, qualitative studies
**Publications**	Published as full-text articles in peer-reviewed scientific journals.	Published as reviews, case reports, conference abstracts, or letters.

**Abbreviations:** Non-RCTs, non-randomised controlled trials; RCTs, randomised controlled trials.

**Table 2 ijms-21-02484-t002:** Searching strategy for the PubMed database.

**("psychiatric disorders"[All Fields] OR "psychiatric disturbances"[All Fields] OR "psychiatric"[All Fields]) AND ("mindfulness"[MeSH Terms] OR "mindfulness"[All Fields] OR mbct[tiab] OR mbsr[tiab] OR "Mindfulness-Based Cognitive Therapy"[tiab] OR "Mindfulness Based Stress Reduction"[tiab] OR "MBI"[tiab] OR "mindfulness-based interventions"[tiab] OR meditation[tiab]) AND ("biomarkers"[MeSH Terms] OR "biomarkers"[All Fields] OR "biomarker"[All Fields] OR "biological markers"[All Fields] OR "biological marker"[All Fields]).**

**Table 3 ijms-21-02484-t003:** Characteristics of studies included in the systematic review.

Study	Population	Age Mean (SD) or IQR	Sex (%F)	Outcomes	MBI Program	Controls	Follow Up (M)	Results	Definition of Psychiatric Disorders	Sd	In
Zgierska	Alcohol	38.4 (8.6)	52.6	IL-6, cortisol	MM	None	4	IL-6 levels decreased from baseline to 16-week follow-up (*p* =	DSM-IV	Open	+
2008 [70]	dependence			serum, salivary	(*n* = 19)			0.052).			+
US					8 weeks			There were no significant changes in salivary cortisol levels			+
								from baseline to 16-week follow-up.			
Gex-Fabry	Remitted from	24–66	71.4	cortisol	MBCT + TAU	TAU	12	No significant changes in cortisol indices from baseline to the	DSM-IV	RCT	+
2012 [65]	recurrent MDD			salivary	(*n* = 28)	(*n* = 28)		end of intervention were observed between both groups.	MADRS ≤13		+
Switzerland					8 weeks	8 weeks					+
Schoenberg	ADHD	19–53	52.3	ERP	MBCT	WL	No	In the MBCT group, there was a significant increase in Pe and	DSM-IV	RCT	?
2014 [66]					(*n* = 24)	(*n* = 20)		NoGo-P3 amplitudes (*p* = 0.02 and *p* = 0.02, respectively).			+
Netherlands					12 weeks						-
Black	Sleep	66.3 (7.4)	67.3	NF-kB	MAPs	Education	No	There was a significant reduction overtime in the levels of NF-	PSQI ≥5	RCT	-
2015 [67]	disturbance			plasma	(*n* = 24)	(*n* = 25)		kB in both groups (*p* = 0.26).			+
US					6 weeks	6 weeks		No significant difference in NF-kB concentrations was observed			-
								between the groups.			
Eisendrath	MDD	34.9 (7.9)	72.7	CRP	MBCT	None	No	There was not significantly decrease in CRP levels from pre- to	SCID DSM-IV	Open	?
2016 [71]				serum	(*n* = 11)			post-intervention (*p* = 0.0517).	HAMD_17_		+
US					8 weeks						-
Walsh	Depressive	19.1 (0.1)	100	IL-6, TNF-α	Mindfulness	Contact	3	Mindfulness training predicted significant decreases in the	CES-D ≥ 16	RCT	+
2016 [68]	symptoms			salivary	(*n* = 31)	Control		levels of IL-6 and TNF-α (*p* < 0.001 and *p* = 0.013, respectively).			+
US					4 weeks	(*n* = 33)4 weeks					-
Memon	MDD, AD or SD	41.5 (11.0)	87.3	IL-6, IL-8, EGF,	Mindfulness	TAU (CBT)	No	EGF levels were significantly decreased from baseline to post-	ICD-10	RCT	-
2017 [23]	and adjustment			hsCRP	(*n* = 81)	(*n* = 85)		intervention in both groups (*p* < 0.001, both).	PHQ-9 ≥10 or		+
Sweden	disorders			plasma	8 weeks	8 weeks		No significant changes in the levels of IL-8 and hsCRP from	HADS-D ≥7		-
								baseline to post-intervention were found in both groups.	or HADS-A ≥7		
									or 13≤ MADRS ≤34		
Wang	MDD, AD or SD	Pt:	Pt:	LTL	Mindfulness	TAU (CBT)	No	At baseline, telomere length was significantly shorter in the	ICD-10	RCT	-
2017 [69]	and adjustment	41.9 (11.1)	87.8		(*n* = 88)	(*n* = 89)		patients compared to the controls (*p* = 0.006).	PHQ-9 ≥10 or		+
Sweden	disorders	Ct:	Ct:		8weeks	8weeks		There were no significant changes in the telomere length from	HADS-D ≥7		-
		44.6 (12.5)	38.4			HCs		baseline to post-intervention in both the Mindfulness and the	or HADS-A ≥7		
						(*n* = 320)		TAU groups, and was no significant difference between the	or 13≤ MADRS ≤34		
								groups.			
Bishop	PTSD	Res:	Res:	*SLC6A4*	MBSR	None	No	There was a significant time x responder group interaction for	DSM-IV	Open	+
2018 [72]		60.4 (14.5)	18.0	*FKBP5*	(Res: *n* = 11)			methylation in *FKBP5* intron 7 bin 2 (*p* = 0.013).	PCL, CAPS		+
US		Non-Res:	Non-Res:		(Non-Res: *n* = 11)			A significant correlation between *FKBP5* intron 7 bin 2			+
		58.2 (10.2)	18.0		9 weeks			methylation change and PCL change from before to after			
								treatment was observed (*r* = −0.451, *p* = 0.04).			
								There was no effect of time for methylation changing in the			
								Primary component of *SLC6A4*.			
Hoge	GAD	39.2 (12.8)	45.7	cortisol, ACTH, IL-6	MBSR	Education	No	MBSR group showed a greater reduction in ACTH, TNF-α and	SCID DSM-IV	RCT	+
2018 [26]				TNF-α during TSST	(*n* = 42)	(*n* = 28)		IL-6 Area Under the Curve (AUC) concentrations compared to			+
US				plasma	8 weeks	8 weeks		control group (*p* =0.007, *p* = 0.033, *p* = 0.036, respectively).			+

**Abbreviations:** SD, standard deviation; IQR, interquartile range; F, female; MBI, mindfulness-based intervention; M, month; Sd, study design; In, intervention quality (Chambless, 1998): low (-) / high ( + ) / unclear (?), from top to down: the study referred to the use of a treatment manual; the therapists who conducted the therapy were trained; treatment integrity was checked during the study; US, United States; IL, interleukin; MM, mindfulness meditation; DSM-IV, Diagnostic and Statistical Manual of Mental Disorders-Fourth Edition; MDD, major depressive disorder; MBCT, mindfulness-based cognitive therapy; TAU, treatment as usual; MADRS, Montgomery Åsberg Depression Rating Scale; RCT, randomised controlled trial; ADHD, attention-deficit hyperactivity disorder; ERP, event-related potential; WL, wait-list; Pe, error-positivity; NF, nuclear factor; MAPs, mindful awareness practices; PSQI, Pittsburgh Sleep Quality Index; CRP, C-reactive protein; SCID, structured clinical interview for DSM-IV; HAMD_17_, Hamilton Depression Rating Scale-17; TNF-α, tumour necrosis factor-α; CES-D, Center for Epidemiologic Studies-Depression Scale; AD, anxiety disorder; SD, stress disorder; EGF, epidermal growth factor; hsCRP, high sensitivity C-reactive protein; CBT, cognitive behavioural therapy; ICD-10, International Statistical Classification of Diseases and Related Health Problems-10; PHQ-9, Patient Health Questionnaire-9; HADS-D, Hospital Anxiety and Depression Scale-Depression; HADS-A, Hospital Anxiety and Depression Scale-Anxiety; Pt, patients; Ct, controls; LTL, Leukocyte telomere length; HCs, healthy controls; PTSD, Post-traumatic stress disorder; Res, responders; PCL, PTSD Checklist; CAPS, Clinician-Administered PTSD Scale; GAD, generalised anxiety disorder; ACTH, adrenocorticotropic hormone; TSST, Trier Social Stress Test; MBSR, mindfulness-based stress reduction.

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
