# Peer review of "Effects of Mindfulness-Based Interventions on Biomarkers and Low-Grade Inflammation in Patients with Psychiatric Disorders: A Meta-Analytic Review"

_ijms, 2020, doi:10.3390/ijms21072484_

Round 1

Reviewer 1 Report

This especially interesting, comprehensive review by Sanada and colleagues is an accurate, well-written and wide-ranged source of current data (in a form of metaanalysis) dealing with potential correlations between mindfulness-based interventions (MBIs) and expression of several stress markers in patients suffered from psychiatric disorders. The topic is very timely and of potential interest for the readers working in the field of psychiatry and neuroscience. It should be underline that relatively large group (about 1000) of participants is undoubtedly an advantage in this work. Nevertheless, authors still understand that further studies are needed to make all conclusions more statistically convincing. It is a proof of their scientific responsibility. Anyway, this one of the best metaanalyses I have ever reviewed.

Author Response

Response to Reviewer #1

We are very honored to receive the comments from the reviewer. Thank you so much.

Reviewer 2 Report

Whilst the number of studies included is low and the variability in the mindfulness-based interventions is quite high, the authors have acknowledged this and included comments in their manuscript. Could the authors clarify whether individual biomarkers (for example TNF) were present across all studies (e.g. GAD and ADHD) and whether they found anything relevant there?

Author Response

Response to Reviewer #2

Thank you very much for the reviewer’s useful comments. We have revised our manuscript accordingly with yellow highlights.

Reviewer #2
1. Could the authors clarify whether individual biomarkers (for example TNF) were present across all studies (e.g. GAD and ADHD) and whether they found anything relevant there?

Response: We appreciate this insightful suggestion, many thanks. We have now added that information in the corresponding “Results” section (p.11, L10-19). Unfortunately, we were not able to develop sub-group meta-analyses taking into account individual biomarkers across studies; for example, a) IL-6 was examined in four studies, but of these studies, one did not include results due to sensitivity issues (Memon et al. 2017) and one did not have a control group (Zgierska et al. 2008); b) cortisol was examined in three studies, but of these studies, one did not have a control group (Zgierska et al. 2008); c) TNF-α was only observed in two studies; and so on. Therefore, we were only able to analyse combined measures of biomarkers in the quantitative synthesis. Nevertheless, we have described the individual results of biomarkers in the “Effectiveness of MBIs on biomarkers” Results section (p.13-14). We have summarized them in the “Discussion” section (p.16) as follows: “MBIs showed statistically significant improvements in the ERP amplitudes in ADHD, in the methylation of serotonin transporter genes in PTSD, in the salivary levels of IL-6 and TNF-α in subjects with depressive symptoms, and in the blood levels of ACTH, IL-6, and TNF-α in GAD.”

Change to Manuscript:

(p.11, L10-19)

The biomarkers included blood IL-6 in patients with depression or anxiety [23], GAD [26], and alcohol dependence [70], salivary IL-6 in subjects with depressive symptoms [68], salivary cortisol in remitted patients with recurrent depression [65] and in patients with alcohol dependence [70], blood cortisol in patients with GAD [26], CRP in patients with MDD [71], high-sensitive CRP (hsCRP) in patients with depression or anxiety [23], blood TNF-α in patients with GAD [26], salivary TNF-α in subjects with depressive symptoms [68], ACTH in patients with GAD [26], EGF in patients with depression or anxiety [23], IL-8 in patients with depression or anxiety [23], leukocyte telomere length in patients with depression or anxiety [69], NF-kB in patients with sleep disturbance [67] and event-related potential (ERP) in patients with ADHD [66].